# Explore the Rare—Molecular Identification and Wine Evaluation of Two Autochthonous Greek Varieties: “Karnachalades” and “Bogialamades”

**DOI:** 10.3390/plants10081556

**Published:** 2021-07-29

**Authors:** Dimitrios Evangelos Miliordos, Georgios Merkouropoulos, Charikleia Kogkou, Spyridon Arseniou, Anastasios Alatzas, Niki Proxenia, Polydefkis Hatzopoulos, Yorgos Kotseridis

**Affiliations:** 1Laboratory of Enology and Alcoholic Drinks, Department of Food Science and Human Nutrition, Agricultural University of Athens, 75 Iera Odos, 11855 Athens, Greece; harakogou@gmail.com (C.K.); spyrosarseniou@gmail.com (S.A.); nprox@aua.gr (N.P.); ykotseridis@aua.gr (Y.K.); 2Department of Vitis, Institute of Olive Tree, Subtropical Crops and Viticulture, Hellenic Agricultural Organisation—DIMITRA, Leoforos Sofokli Venizelou 1, Lykovrysi, 14123 Attiki, Greece; georgios.merkouropoulos@gmail.com; 3Molecular Biology Laboratory, Department of Biotechnology, Agricultural University of Athens, 75 Iera Odos, 11855 Athens, Greece; aalatzas@aua.gr (A.A.); phat@aua.gr (P.H.)

**Keywords:** microsatellite, SSRs, phenolic profile, sensory profile, phenolic compounds, indigenous grapevine varieties, Thrace, Greece

## Abstract

Wines produced from autochthonous *Vitis vinifera* varieties have an essential financial impact on the national economy of Greece. However, scientific data regarding characteristics and quality aspects of these wines is extremely limited. The aim of the current study is to define the molecular profile and to describe chemical and sensory characteristics of the wines produced by two autochthonous red grapevine varieties—“Karnachalades” and “Bogialamades”—grown in the wider area of Soufli (Thrace, Greece). We used seven microsatellites to define the molecular profile of the two varieties, and then we compared their profile to similar molecular data from other autochthonous as well as international varieties. Grape berries were harvested at optimum technological maturity from a commercial vineyard for two consecutive vintages (2017–2018) and vilification was performed using a common vinification protocol: the 2017 vintage provided wines, from both varieties, with greater rates of phenolics and anthocyanins than 2018, whereas regarding the sensory analysis, “Bogialamades” wine provided a richer profile than “Karnachalades”. To our knowledge, this is the first study that couples both molecular profiling and exploration of the enological potential of the rare Greek varieties “Karnachalades” and “Bogialamades”; they represent two promising varieties for the production of red wines in the historic region of Thrace.

## 1. Introduction

Grapevine, one of the oldest and more valuable cultivated agricultural crops [1,2], is cultivated to produce table fruits, raisins, grape juice, and wine. It is estimated that 6000 different grapevine varieties exist around the world although this number is constantly changing due to the creation or disappearance of certain varieties [3]. Modern Eurasian grape cultivars (*Vitis vinifera* ssp. sativa) are originated from the domestication of wild progenitors (*Vitis vinifera* ssp. sylvestris). The domestication event has occurred by multiple wild stocks [4]. Accumulated studies of the last 20 years verified a major impact of local wild grapevines in the development and advancement of current grapevine cultivars in western and central Europe [5].

Viticulture in the region of Thrace (Figure 1) is rooted deeply in antiquity. This area stands as an exceptional case where actually the myth meets the scientific evidence provided by recent archaeological findings. According to the myth, Mount Pangaion, on the south-west part of ancient Thrace, was the place where the god of wine-making, Dionysus, was living together with his followers, the Maenads. Close to mount Pangaion occur the ruins of the Neolithic settlement known as Dikili Tash: archaeological excavation revealed charred grape seeds, grape fruit remains, and two-handed clay cups, suggesting this site as one of the oldest places in Europe where wine production and consumption is evidenced [6].

At the start of the 20th century, cultivation of different grapevine varieties in the valleys of Thrace has been reported [7,8,9]. An extensive destruction in the viticulture production in this area was caused by perpetual conflicts that “converted vineyards into fields” [7] and by the devastating spread of phylloxera, first appearing in Thessaloniki in 1910 [10]. As a consequence, the Greek wine industry was considerably affected during the first half of the 20th century; as a consequence, historic vineyards and grape vine varieties disappeared [11]. Genetic erosion of the autochthonous grapevine germplasm together with the varietal homogenization entailed an increase in genetic vulnerability from the spread of pathogens against susceptible cultivated varieties [12,13]. Reconstruction of the Greek vineyard, in the early 1990s, was based on planting red and white international varieties, such as “Chardonnay”, “Sauvignon blanc”, “Cabernet Sauvignon”, and “Merlot”, which were already acclimatized. The Hellenic Statistical Authority reports confirmed that the main winemaking variety in the region of Thrace, is “Sauvignon blanc” [14]. However, recent guidelines have changed the vineyard planting from international (established) varieties to native stocks. This recaptured new route has also originated from the prominent interest to preserve rare native varieties that are threatened by extinction. Conservation of the local varieties is vital to prevent genetic erosion, and key to recovering remnants of the local cultural heritage. The incorporation of the local rare germplasm within the framework of sustainable management could provide the necessary means to guarantee their survival and to encourage sincere evaluation. Extremely rare reports have been focused on oenological potentials of native cultivars of Greece compared to other international cultivars [15,16]. The trend of “wine regionality” is encouraged by the European Union that favors agricultural production of crops with a clear regional identity retaining safety and quality attributes (EC regulations 2081/92 and 1898/06) [17].

A narrow piece of land of about 20 km in width (Figure 1), within the region of Modern Greek Thrace—the south-west part of ancient Thrace—exhibits significant variation in topography. The climate is strongly influenced by a cool breeze coming from Rhodope Mountains and the warm air from the Aegean Sea. The area around the Evros River (border between Greece and Turkey) has layers of rich alluvial soils with high organic content [18]. Overall, Greek Thrace comprises a broad locality suitable for grapevine cultivation and wine production. Besides the international varieties, a number of local grape varieties with promising attitudes are cultivated in this region while most of them are not identified or characterized. “Karnachalades” and “Bogialamades” varieties have attracted the interest and endeavors of Greek Thracian winemakers for making high-quality red wines with distinct sensorial characteristics.

“Karnachalades” (in plural; Καρναχαλάδες in Greek; [19]) is considered genetically close to the “Bogia”—related varieties (“Bogia”, “Bogiamas”, “Bogialamas”, “Bogialamades”). “Karnachalades” is included in a wine grape guide with “Karnachalas” as its principal synonym [20]. A recent study has shown that “Karnachalades” possess “relatively low degree of genetic similarity” to varieties cultivated in Epirus, the western parts of Greece [21]. “Bogialamades” (in plural; Μπογιαλαμάδες in Greek) is mentioned by Kotinis [19] and Spinthiropoulou [22] as synonym to “Bogiamas” (Μπογιαμάς in Greek). “Bogialamades” is often misinterpreted with the well-known varieties “Alicante Bouschet” and “Grand noir” due to the characteristic attitudes of wines rich in red color; “Bogialamades” was used to color the faint-colored wines produced by “Karnachalades” [23].

In the current research we evaluated the winemaking potential of each of the two red autochthonous Thracian grape varieties “Karnachalades” and “Bogialamades”, and assessed their enological potential using chemical analysis and sensory evaluation. Their molecular profile was defined by seven microsatellites, and was then compared to similar data produced from other Greek and international varieties.

## 2. Results

### 2.1. Molecular Studies

Samples from representative plants of the two varieties, “Karnachalades” and “Bogialamades”, which are cultivated in vineyards around Soufli (Figure 1) were collected. The local empirical names of the collected samples were accepted without prejudice; hence, they are prone to correction. Genomic DNA was isolated from the collected samples, and identified molecularly by the use of seven microsatellites that include all six OIV molecular descriptors [24]. The molecular profiles were then compared to certain Greek and international varieties in order to reveal genetic relationships—preliminary results have been published previously [25].

The DNA amplified fragment allele’s size are shown in Table 1. A total of 29 different DNA amplified fragment alleles were detected using the seven microsatellite loci, with VrZAG67 to be the most polymorphic marker (seven alleles), whereas VVS2, VVMD5, VVMD32, and VrZAG79 markers showed the lowest polymorphism (three alleles).

The molecular profile of “Karnachalades” samples from the Soufli area and the corresponding profile of “Karnachalades” samples from collection maintained by the Hellenic Agricultural Organisation–DIMITRA (ELGO–DIMITRA) at Lykovrysi (Attiki) were identical for the seven microsatellite loci used. The DNA amplified fragment pattern was similar to the profile of the “Karappapas” sample, a variety that is considered autochthonous to Thrace and Epirus [19,26] (Figure 1c). “Bogialamades” of Soufli was found to be almost identical to “Bogialitiko” (collected from Choristi, Drama; Figure 1c), “Grenache” (collected from Grevena; Figure 1c), and “Vapsa” (collected from Elassona; Figure 1c). Noteworthy that the variety “Vapsa”, which is maintained in the ELGO–DIMITRA collection, exhibited a different electrophoretic profile showing difference in at least eight alleles and pointing to a case of homonymy.

The molecular profiles of “Karnachlades” and “Bogialamades” were also compared to similar data of international varieties, pointing out that they represent discrete varieties with distinctive genetic distance to international and domestic germplasm (Figure 2).

### 2.2. Chemical Analysis of Musts

Anthocyanins of the “Karnachalades” and “Bogialamades” grape berries were measured using the Iland method [27] and estimated 0.768 and 4.102 mg/berry in 2017 vintage, and 0.715 and 3.696 mg/berry in 2018 vintage, respectively (Table 2), revealing higher anthocyanins in “Bogialamades” than in “Karnachalades”. Similar trends were recorded for both varieties regarding the total phenols (Table 2). Comparing total phenols and anthocyanins levels, “Bogialamades” grapes had higher values than “Karnachalades”.

Both varieties were harvested earlier in 2018 in comparison to 2017 vintage due to the warmer vegetative season (Figure 3). The °Brix level of “Karnachalades” must in 2018 was higher than in 2017. Similar trends were detected for pH value and berry volume while the opposite tendency was observed for titratable acidity. Comparable results were recorded for the “Bogialamades” musts (Table 3).

### 2.3. Chemical Analysis of Musts

The two varieties produced dry wines (<4 g/L sugar), (Table 3), showing higher values of alcohol level (*v*/*v*) during the 2018 vintage than 2017. Total acidity levels were higher in “Karnachalades” wine during 2017 than in 2018 (Table 4) and the same trend was also observed in “Bogialamades” wine.

Total phenolic index was higher for both “Karnachalades” and “Bogialamades” 2017 wines compared to 2018 (Table 5). Similarly, the Folin–Ciocalteau’s method (Table 5) showed that “Karnachalades” and “Bogialamades” wines produced in 2017 possessed significantly higher levels of total phenolics (1145 and 2926 mg/L gallic acid equivalents, respectively) compared to those produced in 2018 (943 and 2863 mg/L gallic acid equivalents). “Bogialamades” wines showed the highest TPI levels among the two varieties.

“Karnachalades” and “Bogialamades” wines, vintage 2017, scored significantly higher values for both total tannins (as evaluated by MCP index 509 and 1244 tannin mg/L catechin equivalent, respectively) and for astringent tannins (as evaluated by BSA 46.2 and 280.0 mg/L catechin equivalent) compared to those from the 2018 vintage (480.0 and 1102.7 tannin mg/L catechin equivalent, and 40.7 and 250.8 catechin equivalent, respectively) (Table 5).

The method used for anthocyanin determination is based on the effect of pH on anthocyanin structure. Total anthocyanin content was higher in “Bogialamades” wine than “Karnachalades” wine (Table 6). Moreover, the 2017 harvest from both varieties resulted in wines with higher values than that of 2018, was determined as being of 104 and 1106 mg/L, respectively, whereas significantly lower values were observed for 2018 wines, i.e.,: 85 and 1006 mg/L, respectively (Table 6). Similar results were recorded for color intensity levels. Significantly higher values were recorded for 2017 wines (3.2 and 11.2) than for 2018 wines (2.8 and 10.1) of “Karnachalades” and “Bogialamades”, respectively (Table 6).

The content of monomeric anthocyanins in both “Karnachalades” and “Bogialamades” wines, decreased in response to warmer conditions of the year 2018 (Figure 4 and Figure 5). Among the four anthocyanins, malvidin scored significantly the highest concentration in 2017 (59 mg/L for “Karnachalades” and 246 mg/L for “Bogialamades” wine).

In order to highlight the differences observed between the two vintages for both varieties, a PCA was performed on the dataset of variables analyzed in grapes and in their respective wines (Figure 6). The plot obtained for the first PCs is detailed in Figure 5 with PC1 and PC2 explaining, respectively 99.86% of the variation of the data. The PCA analysis allowed an effective separation of both wines produced in 2017 from those produced in 2018. The PCA analysis depicted a distinct partition of “Karnachalades” vs. “Bogialamades”. The biplot distribution clearly separated the wines on the year of production Figure 6. On the left of the Y axis, “Karnachalades” group of wines are located, described by a lower phenolic profile than “Bogialamades”. Grouping of the 2017 and 2018 wines determined in the biplot: the wines of each year were co–located in different parts of the biplot. These results confirmed the effect of the year in the quality characteristics of grapes and wines, showing that warmer vegetative seasons affected both varieties.

### 2.4. Sensory Characteristics

The sensory profile as the outcome of the tasting assessments of 2017 and 2018 “Karnachalades” and “Bogialamades” wines was assessed (Figure 7 and Figure 8). The chemical composition influences the sensory properties of the wine [28]. Color intensity, aroma intensity, aroma of red fruits, and astringency of the red wines produced during those two years were different (Figure 7 and Figure 8). Specifically, the wines produced in 2017 presented higher scores than those produced in 2018, as evaluated for color intensity, aroma intensity, and intensity of fruit aroma by the panelists. On the contrary, the panelists characterized the wines produced in 2018 with higher values for vegetal aroma, a factor that could be associated with the shorter and not balanced ripening period.

## 3. Discussion

In the course of the current project, samples of the Thracian varieties “Karnachalades” and “Bogialamades” were collected from the area around Soufli, in the north–eastern part of Greece. The molecular profile of these samples was then compared to the molecular profile of samples collected either from the ELGO–DIMITRA collection at Lykovrysi or from various sites around the northern parts of Greece; in the latter case, the empirical names of the samples provided by the growers were considered.

The molecular studies showed that the “Karnachalades” samples from Soufli are identical to the variety “Karnachalades” maintained in the ELGO–DIMITRA collection; therefore, they are grouped in the same clade. The “Bogialamades” samples from Soufli are grouped together with: a sample of the international variety “Grenach”, collected from the north–western area of Grevena; two samples of “Vapsa”, one from Elassona and another from the neighboring area of Livadi (Mount Olympos); and one sample of “Bogialitiko”, collected from the north–eastern area of Choristi Drama. It is interesting that these two “Vapsa” samples (from Elassona and Livadi) possess different molecular profile to the “Vapsa” variety that is maintained in the ELGO–DIMITRA collection at Lykovrysi. This discrepancy represents a typical case of homonymy—different genotypes bear the same name—and is under further examination.

The chemical composition of the produced wines is a significant element of the wine characteristics. Especially, the polyphenolic profile of a cultivar reflects to a great extent its genetic and commercial potential, and could be used as a tool to distinguish different cultivars. An important quality characteristic of the red grape berries is their anthocyanin composition. According to Iland method (Table 2), anthocyanins levels of the “Bogialamades” were higher than in “Karnachalades”. Similar trends were recorded for both varieties regarding the total phenols (Table 2). Comparing total phenols and anthocyanins levels, “Bogialamades” grapes had higher values than “Karnachalades”. The grapes coming from “Karnachalades” variety had relatively lower levels of total phenols and anthocyanins than “Agiorgitiko” grapes [29] while grapes from the “Bogialamades” variety had relatively higher total phenols and anthocyanins than “Agiorgitiko”. Gambouti et al. [30] showed that the higher the condensed tannin; the proanthocyanid content of the grapes is the greater the perceived astringency could be. Accordingly, “Bogialamades” grapes could potentially produce more astringent and colored wines.

Both varieties were harvested earlier in 2018 in comparison to 2017 vintage due to the warmer vegetative season (Figure 3). Previous studies have been focused on the effect of temperature in vine physiology and produced wines [31] by associating a number of factors, such as total soluble solids, pH, total acidity and color, to wine quality. Our results confirm that when longer and warmer vegetative seasons take place, a potential increase of the sugar content and pH values, while decrease the total acidity happens [32].

Climate change–related increases of the temperature, pH levels and especially in conjunction with higher sugar/alcohol concentrations, have a direct influence on wine chemistry, like in our study (Table 4; Figure 3).

The phenolic composition can be affected by several factors, such as grape variety, climatic and soil conditions and grape marc maceration time and these contribute significantly to color, taste and the antioxidant properties of wines [33]. The wine samples tested in the present study showed great variations in their phenolic composition indicating great influence especially by the climatic conditions.

According to our finding “Bogialamades” wine recorded higher phenolics and anthocyanins than “Karnachalades” wines (Table 5 and Table 6). Therefore, “Bogialamades” together with another indigenous variety “Thrapsa” [34] could be considered as the red varieties with the highest TPI values among the Greek native grapevines. These values are even higher than the most known Greek red varieties such as “Xinomavro”, “Agiorgitiko”, and “Mandilaria” [35].

It is noteworthy that both varietal 2017 wines scored significantly higher values for total tannins and that the variation found between the phenolic levels in “Bogialamades” wine is 5- to 6-fold higher in comparison to the corresponding levels of the “Karanachaades” wine (Table 5). Hence, the results of MCP and BSA assays could be correlated with the perceived astringency. Comparing the phenolic content to the well–known Greek varieties of “Agiorgitiko”, “Xinomavro” and “Mandilaria” [35], it is evident that “Bogialamades” wine possesses higher phenolic content, whilst “Karnachalades” presents phenolic content lower than “Agiorgitiko” wines. Therefore, “Karnachalades” variety seems that could not lead to the production of wines with ageing potential.

Red color intensity is with no doubt one of the most important characteristics of red wine quality. Color intensity is a significant chromatic characteristic of red wines as they can present a measure of wine color ageing potential and appearance (purple brown for young or old wines, respectively). According to these attributes, “Bogialamades” variety could be used for the production of wines with a considerable ageing potential,

Therefore, the total anthocyanin contend of “Bogialamades” wines showed that it could be considered the richest wine among the two (Table 6), and seemed to have the highest content compared to other Greek red native varieties such as “Mavrothiriko” and “Mavronikolas” with known high total Anthocyanin content [36]. “Bogialamades” wine variety described with a deep red color, while “Karnachalades” shows a lighter red color and could be grouped with the native variety from Ikaria island (South–East Aegean Sea) “Fokiano” [37].

Among the monomeric anthocyanins, according to our results (Figure 4 and Figure 5) malvidin scored the highest concentration. Therefore, malvidin is the dominant anthocyanin in “Karnachalades” and “Bogialamades” wine, followed by peonidin, petunidin, and delphinidin. Similar results were recorded for “Vertzami” Greek rare variety showing high content of monomeric anthocyanins [34]. Warmer seasons during the berry growth are linked with lower values of anthocyanins. Our results confirm previous published data showing significantly lower concentration of anthocyanins in grape berries at the stage of technological maturity in vines grown in pots in 30 °C compared to those grown in 20 °C [38]. It seems that one of the key factors that significantly affect the anthocyanin biosynthesis in the grape skins is the environmental temperature. Accumulated results have shown that temperature influences the anthocyanin grape biosynthesis [38,39,40]. The expression of the anthocyanin biosynthetic genes was induced by low temperature and repressed by high temperature in grapes [39] together with a decrease in expression of flavonoid biosynthetic and MYBA genes in roses [41] and in Arabidopsis [42].

Air temperature was recognized as a key climate variable regulating the environment–genotype relationship in *Vitis vinifera* [43,44], being that the growth habit of this species highly susceptible to thermal regime. Moreover, temperature variability may become the prevalent factor affecting wine’s quality. Unbalanced wines with high alcohol levels, low acidities, modified varietal aroma and a lack of color is typically produced because of elevated temperatures and warmer ripening periods [45]. Moreover, the modification of the secondary metabolite biosynthesis of flavonoids, amino acids and carotenoids could have a direct effect on grape metabolite composition, aroma and flavor [46]. The pH value is also affected by temperature and wines produced during warmer seasons have increased pH values [47].

The climate varies throughout this region during the growing season due to influences of cool breeze from nearby Rhodope mountains. On the other hand, during summer, the coastal areas are not noticeably cooler than inland, since the Aegean is a quite warm sea. Owing to this phenomenon, inland sites accumulate heat to a much greater extent than that of the water, but tend to lose heat rapidly at night. Accordingly, fruit composition and wine quality are strongly affected by heat accumulation during berry set [48]. It is well documented that the phenolic composition of red wines could be influenced by soil, climate, and atmospheric conditions [49,50]. As a consequence, the climate conditions strongly affected the quality characteristics of both studied varieties.

The PCA analysis allowed an effective separation of both wines produced in 2017 from those produced in 2018. The PCA analysis depicted a distinct partition of “Karnachalades” vs. “Bogialamades”. The biplot distribution separated the wines on the year of production Figure 6. On the left of the Y axis, “Karnachalades” group of wines are located, described by a lower phenolic profile than “Bogialamades”. Grouping of the 2017 and 2018 wines determined in the biplot: the wines of each year were co–located in different parts of the biplot. These results confirmed the effect of the year in the quality characteristics of grapes and wines, showing that warmer vegetative seasons affected both varieties.

## 4. Materials and Methods

### 4.1. Plant Material, Genomic DNA Extraction, and Genotyping

Young leaves of grapevine plants cultivated in various productive vineyards in the area around Soufli were collected, immediately frozen in liquid nitrogen, and stored at −80 °C until further treatment. Genomic DNA was extracted from about 100 mg of the frozen tissue using the commercially available NucleoSpin Plant II kit (Macherey–Nagel, Düren, Germany) according to manufacturer’s instructions. The integrity of the extracted genomic DNAs was checked by electrophoresis on agarose gels, while the concentration was estimated by using a Quawell (Q3000 UV–Vis Spectrophotometer, Quawell Technology Inc., San Jose, CA, USA) spectrophotometer.

Polymerase Chain Reactions (PCRs) were performed in a volume of 20 µL using 25 to 30 ng genomic DNA as a template, 200 mM of each dNTP, 10 pmol primers, 4 μL 5X MyTaq Reaction Buffer, and 1u MyTaq DNA Polymerase (Bioline, London, UK). Seven pairs of primers were used: VVS2 [51], VVMD5 and VVMD7 [52], VVMD27 and VVMD32 [53], and VrZAG67 and VrZAG79 [54]. Forward primers were 5′–end fluorescently labeled with different fluorophores: FAM, HEX, ROX and TAMRA. Primers were custom labeled according to each dye’s absorption and emission wavelength and the length of the amplified product in order to avoid overlapping during electrophoresis. PCR amplifications were performed in a 96-well MiniAmp Thermal Cycler (Applied Biosystems, CA, USA) as follows: 1 cycle (95 °C, 2 min), 35 cycles [95 °C, 15 s; 52 to 60 °C (depending on the primers), 15 s; 72 °C, 10 s], and 1 cycle (72 °C, 20 min). PCR fragments were separated using capillary electrophoresis in a 3730xl DNA Analyzer (Applied Biosystems, CA, USA). Data analysis, sizing and genotyping were performed using the GeneMapper (version 4.0) software. GenAlEx 6.5 program [55] was used for statistical analysis. Dendrogram were constructed using the MEGA4 program [56].

### 4.2. Grape Samples

Grape samples, “Karnachalades” and “Bogialamades”, were obtained during the 2017 and 2018 vintages from the commercial “Alania” vineyard (41°17′91.4″ (N) and 26°26′00.3″ (E)), near Soufli region (Figure 1) at technological maturity. Each sample consisted of 60 kg of grapes handpicked and transferred to the experimental winery at the Laboratory of Enology and Alcoholic Drinks at the Agricultural University of Athens, and stored overnight at 4 °C. The following morning, 200 representative berries from each variety were separated for the total anthocyanin and total phenolic content analysis.

### 4.3. Total Anthocyanins and Total Phenolics of Grape Berries

Fifty berries from each treatment were homogenized using Ultra Turrax T25 at 20,000 rpm for 1 min. Total phenolic compounds and anthocyanin content were measured according to Iland et al.; 1 g of the homogenate (in triplicate) was transferred into a centrifuge tube. An amount of 10 mL 50% *v*/*v* aqueous ethanol at pH 2, was then added and mixed for 1 h. After centrifugation at 4000 rpm for 10 min, 0.5 mL of the supernatant was added to 10 mL 1M HC1 and mixed thoroughly. After 3 h, absorbance at 700, 520, and 280 nm was recorded. [27].

### 4.4. Vinification Process

Vinification trials were realized as follows: 60 kg of grapes of each variety were divided in three portions of 20 kgs and then each portion was destemmed and crushed. The produced grape pomaces (musts, skins, and seeds) were placed in 20 liter inox tanks. A total of 10 g/hL of SO2 was added to each tank. Classical musts analyses (pH, °Brix, total acidity) were performed to the musts according to the OIV [57]. Four hours later, inoculation with the Saccharomyces cerevisiae strain SC22 (Fermentis, Lessafre, France) at a rate of 20 g /hL was performed. The skins and juice were mixed three times per day by punching down. The alcoholic fermentation was performed at temperatures between 23 and 25 °C, until sugar depletion. Wines were then separated from skins and seeds, and the pomace was pressed using a hydro fruit press. The pressed wine and free run wine were then mixed and inoculated with Viniflora Oenos (CHR Hansen, Hørsholm, Denmark). Malolactic fermentation (MLF) was considered completed when the malic acid concentration was <0.3 g/L.

### 4.5. Must and Wine Conventional Analyses

Main compositional parameters of grape juice and wines, such as °Brix, residual sugars, pH, titratable acidity, volatile acidity, and alcohol, were determined according to the Compendium of international methods of wine and must analysis [56].

### 4.6. Wine Analysis

Total Phenolic Index (TPI)—Folin−Ciocalteu

Total polyphenol index was measured with a spectrophotometer at 280 nm [58]. Wine samples were diluted and measured at 280 nm. The TPI is calculated as the A280 nm multiplied by the dilution factor.

The total phenolic content was also determined spectrophotometrically at 760 nm, using the Folin–Ciocalteu reagent [59] and was expressed as mg/1 gallic acid (GAE). Briefly, 0.5 mL of FCR is added to 1 mL of a diluted wine (1/5) mixed with 5 mL of distilled water. Then, 2 mL of a 20% (m/v) Na2CO3 solution is added into the mixture. Finally, 2.4 mL of distilled water is incorporated to the test tube. Samples need to be incubated for 30 min before the absorbance is measured. The order of the additions needs to be strictly followed due to the instability of the FCR under alkaline conditions. A sample blank with water instead of wine is used to account for background interference All analyses were performed in triplicate.

### 4.7. Wine Color, Color Density and Total Anthocyanins

According to Sudraud [60] color intensity was determined through molecular absorbance measured at 420, 520 and 620 nm. Total anthocyanins were determined by a spectrophotometric method based on SO2 bleaching [61] and is known as the Bisulfite Bleaching Method. All analyses were performed in triplicate.

### 4.8. Anthocyanins HPLC

Reversed–phase HPLC analyses of anthocyanins were carried out by direct injection of 10 μL of wine into a Waters 2695 Alliance liquid chromatograph system coupled with a Waters 2996 PDA detector (Milford, MA, USA) and using a SVEA C18 Plus 4.6 × 250 mm, 5 μm column (Nanologica, Södertälje, Sweden). The solvents used were 10% aqueous formic acid (solvent A) and methanol (solvent B) in accordance with the method described by Kallithraka et al. [34]. Chromatograms were recorded at 520 nm, and anthocyanin standard curves were made using malvidin–3–*O*–glucoside chloride. Identification was based on comparing retention times of the peaks detected with those of original compounds, and on UV–vis spectrum. The anthocyanidin–3–monoglucosides delphinidin, peonidin, petunidin, and malvidin and the acetylated and p–coumarylated of malvidin were expressed as malvidin equivalents mg/L.

### 4.9. Tannin Determination with Bovine Serrum Albumin and Methyl Cellulose Precipitation

#### 4.9.1. Bovine Serum Albumin Precipitation Assay

One milliliter of BSA solution (1 mg BSA/mL dissolved in a 0.2 M acetic acid and 0.17 M NaCl buffer adjusted to pH 4.9) was added to a 0.5 mL of properly diluted wine. The dilution was made with a model wine solution containing a 12% *v*/*v* EtOH and 5 g/L tartaric acid adjusted to pH 3. After 15 min, the supernatant obtained after centrifugation is discarded. The pellet is then washed twice with 1 mL of the solution adjusted at pH 4.9. After the addition of 0.25 mL of the same solution, the sample is centrifuged for 1 min. The supernatant is again discarded, and the pellet is redissolved by adding 0.875 mL of a TEA buffer at pH 7 or 8. After measurement of the A 510 nm background, 0.125 mL of 10 mM ferric chloride in 10 mM aqueous HCl was added to the sample. The A510 nmFeCl was measured after 10 min. The total tannin content is calculated as follows and expressed as milligrams per liter of catechin equivalents:tannins (mg/L) = A _510 nm_^FeCl 510^ − 0.875 × A _510 nm_ ^background^(1)

Tannin content in this extract was determined according to the protein precipitation method of Harbertson et al. [62,63] Tannin content was quantified against a standard curve of catechin and expressed as milligrams of catechin equivalents per gram fresh weight of skin.

#### 4.9.2. Methylcellulose Precipitation (MCP) Assay

The assay developed and validated by Sarneckis et al. [64]. Briefly, a wine sample is prepared by adding 300 μL of an MCP solution (0.04% *w*/*v*; 1500 cP viscosity at 2%) to 25 μL of wine. After 2−3 min, 200 μL of a saturated solution of (NH4)2SO4 and 475 μL of distilled water are added. A control sample is also prepared, but with distilled water (775 μL) instead of MCP solution. After 10 min, the samples are centrifuged for 5 min, and the tannin content is obtained by comparing the A280 nm control and A280 nm treatment. Tannin concentration was determined by the difference between a control and methylcellulose precipitated sample. Tannin was quantified against a catechin standard curve and expressed as milligrams of catechin equivalents per gram fresh weight of skin. MCP analysis was carried out in triplicates for each of the two red wines, in both vintages (2017 and 2018).

### 4.10. Sensory Analysis

Sensory trials were carried out using 12 trained assessors (equal representation of the two genders) with the majority having professional experience in the wine industry, and with at least two years of experience in red wine tasting. Sensory analysis of wine samples used for the sensory evaluation was performed five months after sealing the tanks of each harvest. The panelists were trained with aroma and taste standards during three sessions. Then, the panelists were trained using red wines in order to acquire homogeneity on the intensity of descriptors. The assessors were recruited from the Laboratory of Enology and Alcoholic Drinks of the Agricultural University of Athens. Samples of wine (25 mL) were presented in a randomized order to each participant. Samples were served in ISO standard glasses 3591 [65] covered with a glass cup (to avoid dispersion of odorants) and coded with a random three–digit number [66]. The evaluation consisted of describing eight descriptors: the tasters proceeded to a quantitative assessment, using a scale from 0 to 9 (lowest to highest intensity) for the following characteristics: color intensity, hue, aroma intensity, red fruits, vegetal aroma, spiciness, taste balance, and astringency.

### 4.11. Statistical Analysis

All values are presented as the mean standard deviation. Statistical analyses were performed using Statgraphics Centurion application (version 1.0.1.C). The significance of the results was determined with an unpaired *t*–test or one–way ANOVA with Tukey’s test. Statistical analyses, namely Principal Component Analysis, and respective graphical representation were performed using XLstat (XLSTAT 2017: Data Analysis and Statistical Solution for Microsoft Excel. Addinsoft, Paris, France, 2017). For the graphs, bars with different symbols were significantly different at * *p* < 0.05.

### 4.12. Meteorological Data

Meteorological data were pooled from the National Observatory of Athens Automatic Network [67]

## 5. Conclusions

Greek vineyards are a vital source of autochthonous varieties and represent valuable genetic resources, although many of these varieties remain unexploited with no viticultural or enological characterization. Rare and endangered grapevine varieties often have desired quality traits and could produce unique regional wines of high quality with high economical potential. Our data provide enological impact for the local and the national winemaking industry that could affect viticulture in this region and lead to exponential commercial use. A detailed rational characterization of the grapevine and the wines produced by minor native grapevines in regions, such as Soufli (Thrace), could be a useful expansion of viticulturists/wineries with positive effects on both local wine industry and territory, and particularly for consumers.

Molecular genetic diversity analysis with standard enological characterization of minor native grapevine varieties, such as “Bogialamades” and “Karnachalades” resulted into the distinctiveness of the variety and their respective produced wines. Phenolic content, anthocyanins concentration and sensory characteristics were varietal– and climate– dependent. This study verified that global warming has been causing shifts in grapevine phenology (budbreak, flowering, harvesting dates) and changing the profile of the produced wine.

Consequently, the current research contributes to autochthonous red varieties and assists to continue further the investigation on rare and minor local varieties. Considering as well, the challenging climate we are facing, these grapevine varieties could be a great source for Greek vineyards that could provide critical remarks concerning vine adaptation in the viticultural–wine industry.

## Figures and Tables

**Figure 1 plants-10-01556-f001:**
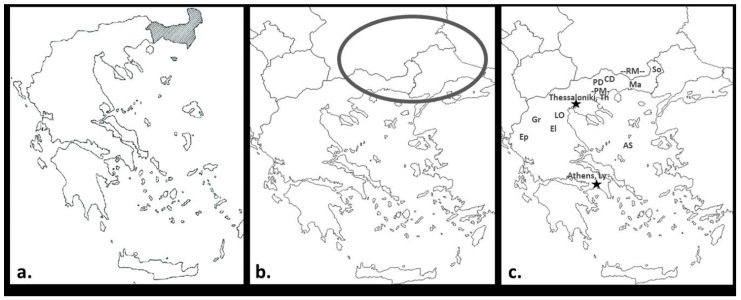
(**a**) Map of (modern) Greek Thrace (grey-colored). (**b**) Area of ancient Thrace (in the black elliptic circle). (**c**) Locations of the main sites mentioned in the manuscript: So: Soufli, Ma: Maronia, RM: Rhodope Mountains, CD: Choristi, Drama, PD Prosotsani, Drama, PM: Mount Pangaion, Th: Thermi, Thessaloniki, LO: Livadi, Mount Olympus, El: Elassona, Gr: Grevena, Ep: Epirus, Ly: Lykovrysi, Athens, AS: Aegean Sea.

**Figure 2 plants-10-01556-f002:**
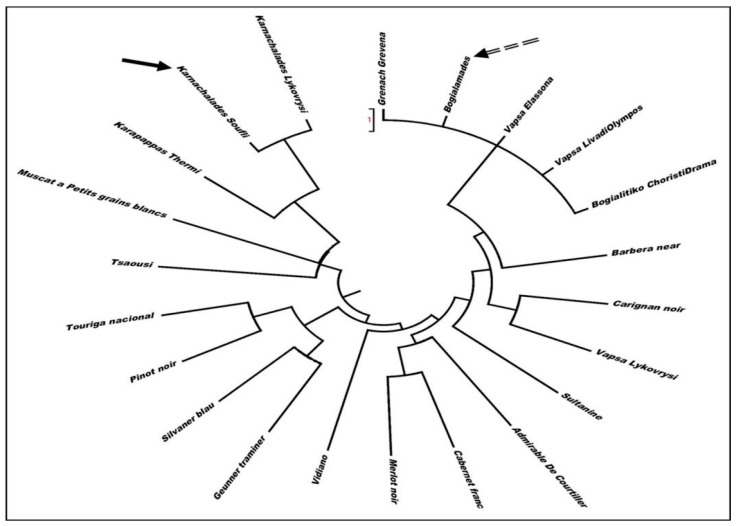
UPGMA dendrogram showing genetic relationship of the Soufli varieties to the international ones.

**Figure 3 plants-10-01556-f003:**
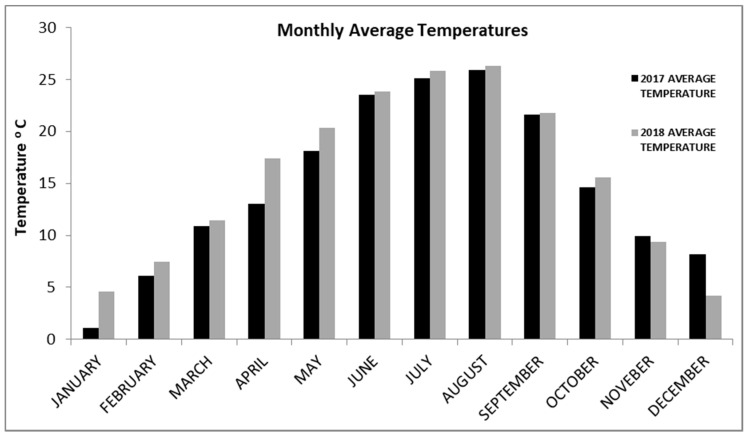
Monthly average temperatures of Soufli area during 2017 and 2018.

**Figure 4 plants-10-01556-f004:**
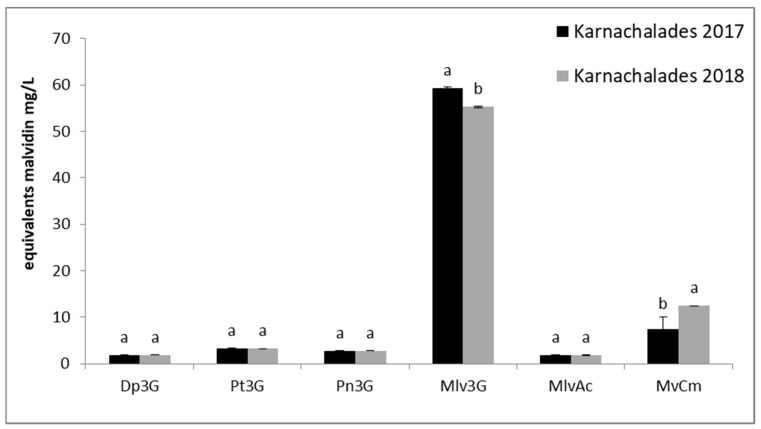
Individual anthocyanins evaluated by HPLC in “Karnachalades” (a) wines. Abbreviations: Dp3G, delphinidin–3–*O*–glucoside; Pt3G, petunidin–3–*O*–glucoside; Pn3G, peonidin–3–*O*–glucoside; Mlv3G, malvidin–3–*O*–glucoside; MlvAc, malvidin 3–*O*–acetate–glucoside; MvCm, malvidin 3–*O*–coumarate–glucoside. Error bars show the average standard error. Different letters indicate significant difference between the same compounds in two consecutive years at *p* < 0.05 according to *t*-test.

**Figure 5 plants-10-01556-f005:**
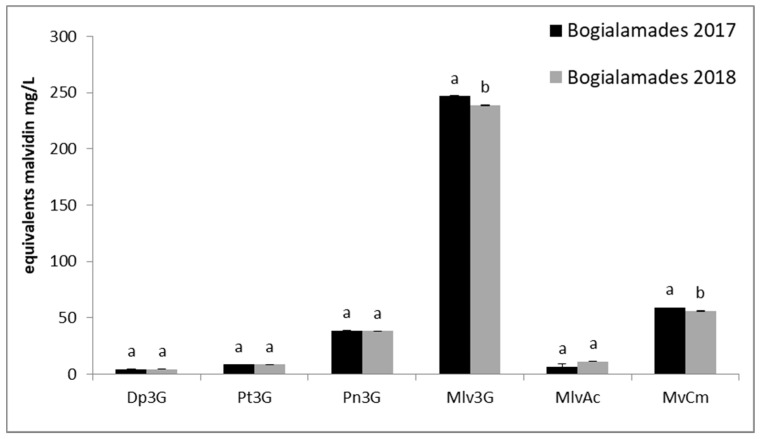
Individual anthocyanins evaluated by HPLC in “Bogialamades” wines. Abbreviations: Dp3G, delphinidin–3–*O*–glucoside; Pt3G, petunidin–3–*O*–glucoside; Pn3G, peonidin–3–*O*–glucoside; Mlv3G, malvidin–3–*O*–glucoside; MlvAc, malvidin 3–*O*–acetate–glucoside; MvCm, malvidin 3–*O*–coumarate–glucoside. Error bars show the average standard error. Different letters indicate significant difference between the same compounds in two consecutive years at *p* < 0.05 according to *t*-test.

**Figure 6 plants-10-01556-f006:**
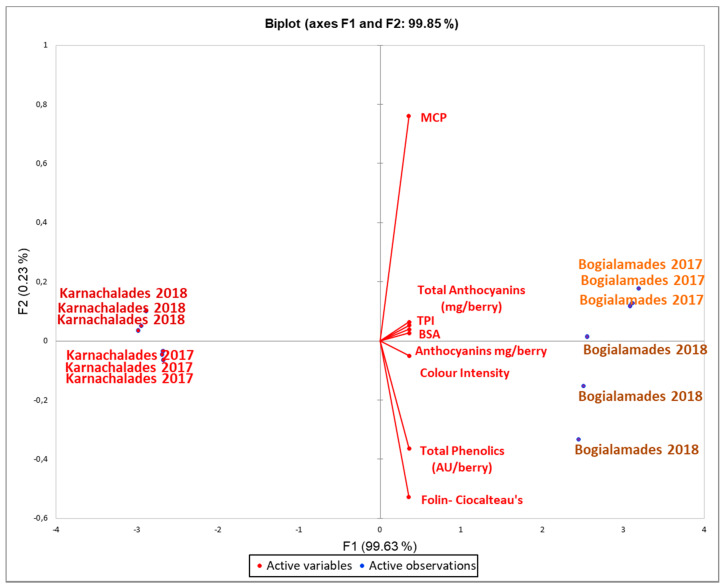
Biplot (PCA) for anthocyanins, total phenolics in grape berries and for color intensity, total anthocyanins, MCP, Folin–Ciocalteau’s, BSA and MCP method, TPI in the wines. Samples are grouped following the cultivar and the vintage. Left and bottom axes of the PCA plot are PCA scores of the samples (dots). Top and right axes of the PCA plot indicate how strongly each characteristic (vector) influence the principal components. PC1 and PC2 explaining, respectively, 99.86% of the variation of the data. Variables in plot were colored according to the variety.

**Figure 7 plants-10-01556-f007:**
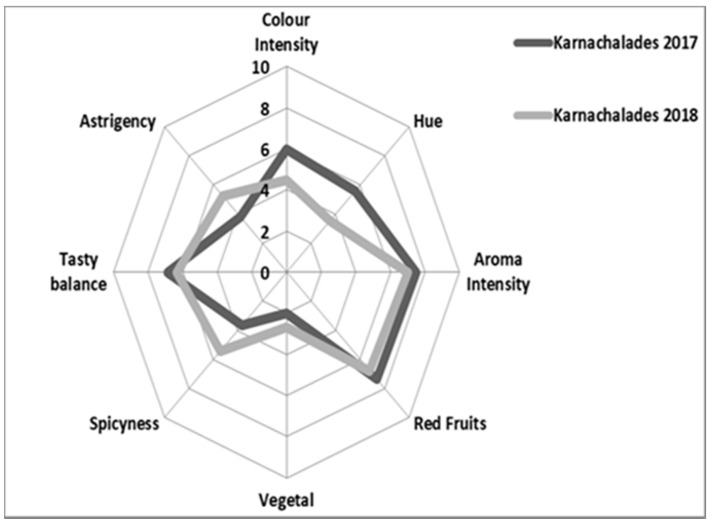
Sensory profile (Spider Diagram) of “Karnachalades” wines in two consecutive years, outlined by a group of professional tasters from the Laboratory of Enology and Alcoholic Beverages of the Agricultural University of Athens. The centers of the diagrams represent the lower values with the tension of each attribute increasing to an intensity of the digit 10 at the perimeter.

**Figure 8 plants-10-01556-f008:**
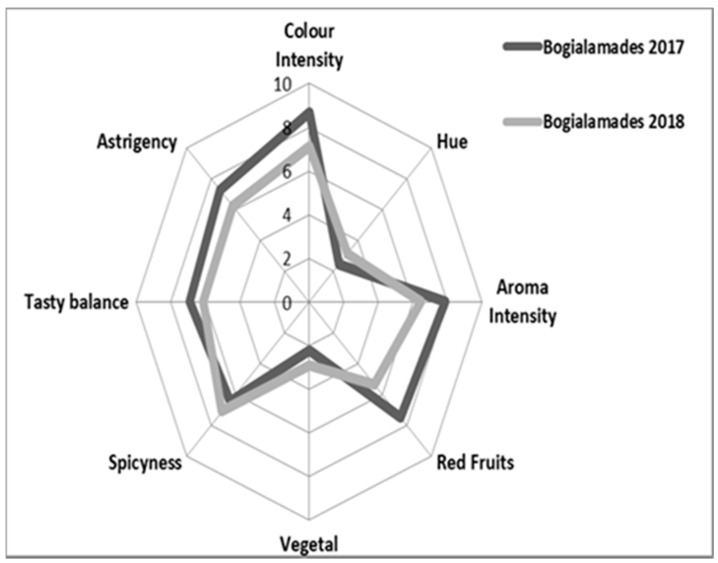
Sensory profile (Spider Diagram) of “Bogialamades” wines in two consecutive years, outlined by a group of professional tasters from the Laboratory of Enology and Alcoholic Beverages of the Agricultural University of Athens. The centers of the diagrams represent the lower values with the tension of each attribute increasing to an intensity of the digit 10 at the perimeter.

**Table 1 plants-10-01556-t001:** Allele sizes (bp) of 7 microsatellite loci from 10 grapevine varieties.

Variety *	Allele Sizes (bp) of 7 Microsatellite Loci from 10 Grapevine Varieties						
	VvS2	VvMD5	VvMD7	VvMD27	VvMD32	VvZAG67	VvZAG79
Karapappas–T	131–143	223–223	246–248	183–191	270–270	152–158	247–247
Karnachalades–S	141–143	223–225	248–248	179–183	270–270	150–158	241–247
Karnachalades–L	141–143	223–225	248–248	179–183	270–270	150–158	241–247
Bogialitiko–DP	131–143	223–235	238–242	179–191	248–270	130–138	241–255
Unknown–DC	131–143	223–235	238–242	179–191	248–270	130–138	241–255
Vapsa–LO	131–143	223–235	238–242	179–191	248–270	130–138	241–255
Vapsa–E	131–143	223–235	238–242	179–191	248–270	130–138	241–255
Vapsa–L	141–143	223–223	232–238	177–179	248–260	122–136	247–255
Grenach–G	131–143	223–235	238–242	179–191	248–270	130–138	241–255
Bogialamades–S	131–143	223–235	238–242	179–191	248–270	130–136	241–255

*: DC = Choristi, Drama; DP = Prosotsani, Drama; E = Elassona; G = Grevena; L = Lykovrysi, Attiki; LO = Livadi, Mt Olympus; S = Soufli, Thrace; T = Thermi, Thessaloniki.

**Table 2 plants-10-01556-t002:** Results from One–Way Analysis of Variance (ANOVA) with Tukey’s test, between total phenols and total anthocyanins in grape berries according to Iland method.

Variety	Vintage	Total Phenols (AU/Berry)	Anthocyanins (mg/Berry)
Karnachalades	2017	1.702 ± 0.027 a	0.768 ± 0.017 a
Karnachalades	2018	1.565 ± 0.009 b	0.715 ± 0.012 b
Bogialamades	2017	3.596 ± 0.001 a	4.102 ± 0.099 a
Bogialamades	2018	3.523 ± 0.011 b	3.696 ± 0.077 b

Mean ± St. Deviation (St. Dev.) followed by the same letter, in the column, do not differ by *t*-test at 5% probability.

**Table 3 plants-10-01556-t003:** Results from One–Way Analysis of Variance (ANOVA) with Tukey’s test of the conventional wine analysis parameters of the “Karnachalades” and “Bogialamades” must in two consecutive vintages.

Variety	Harvest Time	Berry Volume (mL/Berry)	Total Soluble Solids (°Brix)	Tot. Acid. (g/L Tart. Ac.)	pH
Karnachalades	19/10/2017	2.6 ± 0.02 b	20.9 ± 0.03 b	4.1 ± 0.06 a	3.53 ± 0.04 b
Karnachalades	10/10/2018	2.8 ± 0.04 a	22.5 ± 0.04 a	3.9 ± 0.05 b	3.75 ± 0.08 a
Bogialamades	19/10/2017	2.2 ± 0.07 b	19.8 ± 0.06 b	4.7 ± 0.04 a	3.29 ± 0.08 b
Bogialamades	10/10/2018	2.4 ± 0.06 a	23.1 ± 0.08 a	3.8 ± 0.04 b	3.82 ± 0.09 a

Data represent Mean ± St. Deviation (St. Dev.) followed by the same letter, in the column, do not differ by *t*-test at 5% probability.

**Table 4 plants-10-01556-t004:** Results from One–Way Analysis of Variance (ANOVA) with Tukey’s test, of the conventional wine analysis parameters (alcohol strength, pH, total acidity,) of “Karnachalades” and “Bogialamades” wines in two consecutive vintages.

Variety	Year	Ethanol (*v*/*v* %)	Total Acidity (Tar. Ac. g/L)	Volatile Acidity (Ac. Ac. g/L)	Residual Sugars (g/L)	pH
Karnachalades	2017	11.2 ± 0.8 b	5.32 ± 0.08 a	0.40 ± 0.06 a	1.79 ± 0.15 b	3.58 ± 0.12 b
Karnachalades	2018	13.5 ± 0.4 a	5.17 ± 0.11 b	0.36 ± 0.05 a	2.32 ± 0.05 a	3.88 ± 0.07 a
Bogialamades	2017	11.2 ± 0.5 b	5.3 ± 0.05 a	0.36 ± 0.09 a	2.32 ± 0.12 a	3.63 ± 0.10 b
Bogialamades	2018	14.3 ± 0.7 a	5.17 ± 0.04 b	0.23 ± 0.08 a	1.98 ± 0.14 b	3.88 ± 0.07 a

Data represent Mean ± St. Deviation (St. Dev.) followed by the same letter, in the column, do not differ by *t*-test at 5% probability.

**Table 5 plants-10-01556-t005:** Results from One–Way Analysis of Variance (ANOVA) with Tukey’s test, of phenolic characteristics of “Karnachalades” and “Bogialamades” wines in two consecutive vintages.

Variety	Year	TPI	Folin–Ciocalteau’s (Gal. Ac. mg/L)	MCP (Catechin mg/L)	BSA (Catechin mg/L)
Karnachalades	2017	26.1 ± 0.4 a	1145 ± 8 a	509 ± 4 a	46.2 ± 0.9 a
Karnachalades	2018	23.6 ± 1.1 b	943 ± 7 b	480 ± 4 b	40.6 ± 1.2 b
Bogialamades	2017	92.3 ± 1.5 a	2926 ± 12 a	1244 ± 13 a	280.0 ± 1.3 a
Bogialamades	2018	84.5 ± 0.9 b	2863 ± 5 b	1102 ± 76 b	250.8 ± 4.0 b

Data represent Mean ± St. Deviation (St. Dev.) followed by the same letter, in the column, do not differ by *t*-test at 5% probability.

**Table 6 plants-10-01556-t006:** Results from One–Way Analysis of Variance (ANOVA) with Tukey’s test, of color characteristics of “Karnachalades” and “Bogialamades” wines in two consecutive vintages. Mean ± St. Dev.

Variety	Year	Color Intensity	Total Anthocyanins (mg/L)
Karnachalades	2017	3.2 ± 0.1 a	104 ± 4 a
Karnachalades	2018	2.8 ± 0.1 b	85 ± 5 b
Bogialamades	2017	11.2 ± 0.3 a	1106 ± 25 a
Bogialamades	2018	10.1 ± 0.3 b	1007 ± 25 b

Data represent Mean ± St. Deviation (St. Dev.) followed by the same letter, in the column, do not differ by *t*–test at 5% probability.

## Data Availability

The data presented in this study are available on request from the corresponding author (pending privacy and ethical considerations).

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
