# Peer review of "Explore the Rare—Molecular Identification and Wine Evaluation of Two Autochthonous Greek Varieties: “Karnachalades” and “Bogialamades”"

_plants, 2021, doi:10.3390/plants10081556_

Round 1

Reviewer 1 Report

The topic is quite interesting. Before the final decision, I suggest doing minor changes as are pointed out below.

Authors should work on the quality of figures because the text is hard to read. It seems that the content of the figure is very squeezed.

Also, the number of chapters in Materials and methods should be reorganized. for example: "Total anthocyanins and total phenolics of grape berries" should be 4.2.1.

Figure 5. misspelling in anthocyjanins

Figure 4. the description below the figure has to be checked.

Table 6. the units for colour intensity should be added.

Line 286 the sentence needs to be corrected.

I would rather prefer to see the type of analysis than "Results from One-Way Analysis of Variance (ANOVA)".

I disagree with the statement, that biplot shows "an effective separation of both wines produced in 2017 from those produced in 2018". It shows differences between wine varieties, but results from 2017 and 2018 are very close. This conclusion should be once re-written. 

The DOI number needs to be added to the references. Moreover, the cited literature is not very up-to-date a lot of articles are older than 10 years. I suggest exchanging the literature with articles not older than 5-10 years. 

Author Response

Response to Reviewer 1 Comments

Comments and Suggestions for Authors

The topic is quite interesting. Before the final decision, I suggest doing minor changes as are pointed out below.

Authors should work on the quality of figures because the text is hard to read. It seems that the content of the figure is very squeezed

We thank the reviewer for pointing that interesting issue. In the revised manuscript we have re- arranged the figures in order to be clear and read by the readers. The figure 4(a & b ) (Anthocyanins by HPLC ) and Figure 6(a & b) (Spider diagrams for sensory analysis) were removed from the two columns and become larger the figures. Therefore, Figure 4a turned to 4, 4b to 5 and 6a to 7 and 6b to 8 with the appropriate description above every figure.

Also, the number of chapters in Materials and methods should be reorganized. for example: "Total anthocyanins and total phenolics of grape berries" should be 4.2.1.

We thank the reviewer for pointing that mistake. In the revised manuscript the chapters have been re- organized and some paragraphs were re- written

Figure 5. misspelling in anthocyjanins

We thank the reviewer for pointing out that misspelling. We have reviewed and it is corrected.

Figure 4. the description below the figure has to be checked.

We thank the reviewer for pointing out this issue. Therefore we re- wrote the description below the figure into “Figure 4b: Individual anthocyanins evaluated by HPLC in “Bogialamades”wines. Abbreviations: Dp3G, delphinidin-3-O-glucoside; Pt3G, petunidin-3-O-glucoside; Pn3G, peonidin-3-O-glucoside; Mlv3G, malvidin-3-O-glucoside; MlvAc, malvidin 3-O-acetate-glucoside; MvCm, malvidin 3-O-coumarate-glucoside. Error bars show the average standard error. Different letters indicate significant difference between the same compounds in two consecutive years at p < 0.05 according to t- Test.”

Table 6. the units for colour intensity should be added.

An important measures of wine quality can be evaluated by mathematical combination of absorbance values at multiple wavelengths. According to OIV-MA-BS-26(Compendium of international methods of analysis of wines and musts) Wine Color Intensity – a simple measure of how dark the wine is using a summation of absorbance measurements in the violet, green and red areas of the visible spectrum. Wine color intensity = A420 + A520 + A620. Therefore, due to the fact that this value is an index, in the field of Oenology doesn’t use unit

Line 286 the sentence needs to be corrected.

We thank the reviewer for pointing that interesting issue. In the revised manuscript we have corrected the sentence and replace with the “The grapes coming from “Karnachalades” grapes variety had relatively lower levels of Ttotal Phenols and Anthocyanins than “Agiorgitiko” grapes [29] but while grapes from “Bogialamades” variety had relatively higher Total Phenols and Anthocyanins than “Agiorgitiko”.

I would rather prefer to see the type of analysis than "Results from One-Way Analysis of Variance (ANOVA)".

We thank the reviewer for pointing that interesting issue. As a consequence we replaced it with the sentence “Results from One-Way Analysis of Variance (ANOVA) with Tukey’s test” adding the test that have been used for the analysis

I disagree with the statement, that biplot shows "an effective separation of both wines produced in 2017 from those produced in 2018". It shows differences between wine varieties, but results from 2017 and 2018 are very close. This conclusion should be once re-written.

Principal component analysis (PCA) is a statistical technique that provides a linear transformation of an original set of variables into a substantially smaller set of uncorrelated variables, called principal components (PC). PCA represents most of the information in the original set of variables. Much of the variability in the original data can be accounted for by a rst few PC. A smaller set of PC is easier to understand than a large number of original variables. PCA can be used as a linear predictor in the identification of patterns in complex data sets such. Moreover, PCA often generates components that have valuable biological meanings. In the current research from the generated PCA biplot there is a clear separation of the two wine varieties, “Karnachalades” and “Bogialamades”, with the y axe. At the left side of the biplot is grouped the “Karnachalades” wine variety of the two vintages and at the right side is grouped the “Bogialamades” wine variety of the two vintages. In addition to that, there is a separation as well, between the two consecutive vintages with the x axe. However, as it is mentioned by the results from 2017 and 2018 are very close, we erased the words from the text that shows the strong separation, like “strongly” and “clear separation”.

In addition to that the separation between the two vintages could be seen as well from the ANOVA that has been performed in the two varieties between the vintages and we observed that there are significant differences at the chemical composition of the wines and in the quality characteristics such as color, tannin and anthocyanins.

The DOI number needs to be added to the references. Moreover, the cited literature is not very up-to-date a lot of articles are older than 10 years. I suggest exchanging the literature with articles not older than 5-10 years.

We thank the reviewer for pointing the mistake for not writing the DOI numbers in the references.

What concerns the “age” of the articles which are being cited, In present scientific manuscript, references are the information that are necessary to the reader in identifying and finding used sources

Sources in the present manuscript include as well a few history paper (7, 8, 9, 10 and 18), created long ago and used as primary sources in order to provide to the readers the cultural view of the area and the correlated with the viticulture of the specific area. We strongly believe that those “older papers” can be informative because they contain a lot more details and they describe the situation of the viticultural heritage of the wider area of Greek- Thrace. Moreover, in the current manuscript it is the first time that this area is presented to be of viticultural importance. Therefore, those older paper could be included.

On the other hand an older reference that is often cited, even more than 500 times, often in the articles we are using in field of molecular identification (51, 52, 53, 54) and in Oenology (57, 58, 59 ) are the reference points and we believe it’s worth including them, and cite them as a primary sources.

However, a reference which was old enough [61] (Riberau Gayon, 1965) has been erased and replaced with the McDougall et al. 2005

Reviewer 2 Report

The manuscript entitled “Explore the Rare – Molecular Identification and Wine Evaluation of two Autochthonous Greek Varieties: “Karnachalades” and “Bogialamades” “ describes the chemical composition of grape and wine of two Greek varieties. The manuscript is well structured, and all conducted experiments are according to high scientific standards. However, before publication, some changes must be done.

Lines 34,64 – links for websites should be added to the Reference list together with the accession dates

Line 37 – please change V. vinifera to Vitis vinifera

Line 61 – all varieties should be written with a quote, thus please change Chardonnay to “Chardonnay” etc.

Line 176 – please change v/v to v/v (italic)

Table 4. – please change V/V to v/v

Line 186-187 – please check units for Total phenolics – all other parameters are expressed in mg/L

Figure 4. – Authors didn’t perform determination of anthocyanidins (aglycons) so please change delphinidin and other compounds to the appropriate glucosides

Line 339-340, 377 – please correct signs for degree Celsius

Sub-sections 4.3., 4.7., and 4.10. are poorly written. Missing some crucial information about used methods. Please provide more details.

Line 446-448 – pleased provide information about identification of the listed anthocyanins

Author Response

REVIEWER 2

Comments and Suggestions for Authors

The manuscript entitled “Explore the Rare – Molecular Identification and Wine Evaluation of two Autochthonous Greek Varieties: “Karnachalades” and “Bogialamades” “ describes the chemical composition of grape and wine of two Greek varieties. The manuscript is well structured, and all conducted experiments are according to high scientific standards. However, before publication, some changes must be done.

Lines 34,64 – links for websites should be added to the Reference list together with the accession dates

We thank the reviewer for pointing that issue. Therefore, we erase the links of the websites, in the text and we added the reference. For instance in L64 we erased the website link and added the reference [14] and in figure 3 we erased the webside and added the reference as well.

Line 37 – please change V. vinifera to Vitis vinifera

We thank the reviewer for pointing that mistake. Therefore we corrected as the reviewer suggested.

Line 61 – all varieties should be written with a quote, thus please change Chardonnay to “Chardonnay” etc.

We thank the reviewer for pointing that issue. Therefore we corrected as the reviewer suggested all the varieties written in the text.

Line 176 – please change v/v to v/v (italic)

We thank the reviewer for pointing that issue. Therefore we corrected as the reviewer suggested

Table 4. – please change V/V to v/v

We thank the reviewer for pointing that issue. Therefore we corrected as the reviewer suggested

Line 186-187 – please check units for Total phenolics – all other parameters are expressed in mg/L

We thank the reviewer for pointing that issue. Therefore we corrected as the reviewer suggested. Therefore, it is written “Figure 4b: Individual anthocyanins evaluated by HPLC in “Bogialamades”wines. Abbreviations: Dp3G, delphinidin-3-O-glucoside; Pt3G, petunidin-3-O-glucoside; Pn3G, peonidin-3-O-glucoside; Mlv3G, malvidin-3-O-glucoside; MlvAc, malvidin 3-O-acetate-glucoside; MvCm, malvidin 3-O-coumarate-glucoside. Error bars show the average standard error. Different letters indicate significant difference between the same compounds in two consecutive years at p < 0.05 according to t- Test.”

Figure 4. – Authors didn’t perform determination of anthocyanidins (aglycons) so please change delphinidin and other compounds to the appropriate glucosides

We thank the reviewer for pointing that issue. Therefore we corrected as the reviewer suggested

Line 339-340, 377 – please correct signs for degree Celsius

We thank the reviewer for pointing that issue. Therefore we corrected as the reviewer suggested

Sub-sections 4.3., 4.7., and 4.10. are poorly written. Missing some crucial information about used methods. Please provide more details.

We thank the reviewer for pointing that. As a consequence we revised the subsection and re- written in details and re- arranged in correct subsection in the “Materials and Methods” . For instance

4.3 Total anthocyanins and total phenolics of grape berries

Fifty berries from each treatment were homogenized using Ultra Turrax T25 at 20,000 rpm for 1 min. Total phenolic compounds and anthocyanin content were measured according to Iland et al; 1 g of the homogenate (in triplicate) was transferred into a centrifuge tube. An amount of 10 mL 50% v/v aqueous ethanol at pH 2, was then added and mixed for 1 h. After centrifugation at 4,000 rpm for 10 min, 0.5 mL of the supernatant was added to 10 mL 1M HC1 and mixed thoroughly. After 3 h, absorbance at 700, 520, and 280 nm was recorded. [27].

4.6 Wine Analysis

Total Phenolic Index (TPI) - Folin−Ciocalteu

Total polyphenol index was measured with a spectrophotometer at 280 nm [58]. Wine samples were diluted and measured at 280 nm. The TPI is calculated as the A280 nm multiplied by the dilution factor

The total phenolic content was also determined spectrophotometrically at 760 nm, using the Folin-Ciocalteu reagent [59] and was expressed as mg/1 gallic acid (GAE). Briefly, 0.5 mL of FCR is added to 1 mL of a diluted wine (1/5) mixed with 5 mL of distilled water. Then 2 mL of a 20% (m/v) Na2CO3 solution is added into the mixture. Finally, 2.4 mL of distilled water is incorporated to the test tube. Samples need to be incubated for 30 min before the absorbance is measured. The order of the additions needs to be strictly followed due to the instability of the FCR under alkaline conditions. A sample blank with water instead of wine is used to account for background interference. All analyses were performed in triplicate.

AND

4.9 Tannin determination with Bovine Serrum Albumin and Methyl Cellulose Precipitation

Bovine Serrum Albumin precipitation assay

One milliliter of BSA solution (1 mg BSA/mL dissolved in a 0.2 M acetic acid and 0.17 M NaCl buffer adjusted to pH 4.9) was added to a 0.5 mL of properly diluted wine. The dilution was made with a model wine solution containing a 12% v/v EtOH and 5 g/L tartaric acid adjusted to pH 3. After 15 min, the supernatant obtained after centrifugation is discarded. The pellet is then washed twice with 1 mL of the solution adjusted at pH 4.9. After the addition of 0.25 mL of the same solution, the sample is centrifuged for 1 min. The supernatant is again discarded, and the pellet is redissolved by adding 0.875 mL of a TEA buffer at pH 7 or 8. After measurement of the A 510 nm background, 0.125 mL of 10 mM ferric chloride in 10 mM aqueous HCl was added to the sample. The A510 nmFeCl was measured after 10 min. The total tannin content is calculated as follows and expressed as milligrams per liter of catechin equivalents:

tannins (mg/L) = A 510 nmFeCl 510 – 0,875× A 510 nm background

Tannin content in this extract was determined according to the protein precipitation method of Harbertson et al. [62, 63] Tannin content was quantified against a standard curve of catechin and expressed as milligrams of catechin equivalents per gram fresh weight of skin.

Methylcellulose precipitation (MCP) assay

The assay developed and validated by Sarneckis et al. [64]. Briefly a wine sample is prepared by adding 300 μL of a MCP solution (0.04% w/v; 1500 cP viscosity at 2%) to 25 μL of wine. After 2−3 min, 200 μL of a saturated solution of (NH4)2SO4 and 475 μL of distilled water are added. A control sample is also prepared, but with distilled water (775 μL) instead of MCP solution. After 10 min, the samples are centrifuged for 5 min, and the tannin content is obtained by comparing the A280 nmcontrol and A280 nmtreatment. Tannin concentration was determined by the difference between a control and methyl-cellulose precipitated sample. Tannin was quantified against a catechin standard curve and expressed as milligrams of catechin equivalents per gram fresh weight of skin.

MCP analysis was carried out in triplicates for each of the two red wines, in both vintages (2017 and 2018).

Line 446-448 – pleased provide information about identification of the listed anthocyanins

We thank the reviewer for pointing that. Therefore, we revised the subsection and re- written in details.

Reversed-phase HPLC analyses of anthocyanins were carried out by direct injection of 10 μl of wine into a Waters 2695 Alliance liquid chromatograph system coupled with a Waters 2996 PDA detector (Milford MA, USA) and using a SVEA C18 Plus 4.6X250 mm, 5 μm column (Nanologica, Södertälje, Sweden).  The solvents used were 10% aqueous formic acid (solvent A) and methanol (solvent B) in accordance with the method described by Kallithraka et al. [34]. Chromatograms were recorded at 520 nm, and anthocyanin standard curves were made using malvidin-3-O-glucoside chloride. Identification was based on comparing retention times of the peaks detected with those of original compounds, and on UV–vis spectrum. The anthocyanidin-3-monoglucosides del-phinidin, peonidin, petunidin, and malvidin and the acetylated and p-coumarylated of malvidin expressed as malvidin equivalnts mg/L.
